# The Influence of Unlimited Sucrose Intake on Body Weight and Behavior—Findings from a Mouse Model

**DOI:** 10.3390/brainsci12101332

**Published:** 2022-09-30

**Authors:** Olga Dubljević, Vanja Ković, Željko Pavković, Miloš Mitić, Vesna Pešić

**Affiliations:** 1Department of Neurobiology, Institute for Biological Research ‘‘Siniša Stanković’’—National Institute of Republic of Serbia, University of Belgrade, Bulevar Despota Stefana 142, 11060 Belgrade, Serbia; 2Laboratory for Neurocognition and Applied Cognition, Department of Psychology, Faculty of Philosophy, University of Belgrade, 11000 Belgrade, Serbia; 3Department of Molecular Biology and Endocrinology, ‘‘VINČA’’ Institute of Nuclear Sciences—National Institute of the Republic of Serbia, 11000 Belgrade, Serbia

**Keywords:** anxiety, social dominance, novelty, exploration, C57BL/6J mice

## Abstract

A potential relationship between unrestricted sucrose intake (USI), overweight, and emotional/behavioral control has not been well documented. We examined the influence of USI and having less sweetness than expected on body weight (BW), motor/exploratory, anxiety-like, and social dominant behavior in adult C57BL/6J male mice. Animals had free access to water (group 1) or 32% sucrose and water (sucrose groups 2–5) for 10 days. Then, group 2 remained with 32% sucrose while groups 3–5 were subjected to the downshift (24 h access to 4%, 8%, or 16% sucrose). All experimental groups were weighed and tested in the novel-open arena (NA), elevated plus maze (EPM), and tube tests to assess BW, motor/exploratory, anxiety-like, and social dominance behavior, respectively. USI did not influence animals’ BW but produced hyperactivity and anxiolytic-like behavior, which was evident in EPM but not in NA; the outcomes of the downshift were comparable. USI did not influence successes/wins in the tube test but altered emotions that drive the winning, favoring a less anxious behavioral phenotype; this was not evident in the downshifted groups. Observed findings suggest that USI promotes sensation-seeking and motivates dominance, without changing BW, while blunted emotional base of social dominance might be an early mark of the downshift.

## 1. Introduction

Direct and indirect relationships between the consumption of refined sugars and chronic disease incidence, such as insulin resistance, fatty liver, cardiovascular disease, metabolic syndrome, visceral adiposity, hyperuricemia, and type 2 diabetes, have been well documented thus far in literature [1]. In addition to these well-known disadvantages to a sugary diet, there is growing evidence that sucrose (i.e., table sugar) has similar addictive properties as common drugs of abuse [2] since, when ingested intermittently, it causes recurrent increases in extracellular dopamine level in areas of the brain that process behavior reinforcement in a manner that is more characteristic for certain drugs of abuse (such as cocaine) than palatable foods (for review see [3]). This effect is absent when sugar is present ad libitum, as it loses novelty and does not produce bingeing, i.e., the escalation of intake with a high proportion of intake at one time [3]. Consequently, unlike metabolic and obesity-related health disparities, behavioral outcomes of unlimited sucrose intake (USI) outside the food context have not gotten much attention in experimental research, even though this manner of sugar intake is a more valid representation of human daily exposure to sucrose. 

A potential contribution of USI to the formation of certain behavioral phenotypes, especially those associated with externalized behavior (disinhibition, antisocial-aggressive behavior, and substance (mis)use [4]), has been attracting the attention of the scientific public lately, considering that, on one side, unrestricted consumption of high-sugar food and beverages might be linked to the increased obesity epidemic [5] while, on the other side, the association between attention-deficit/hyperactivity disorders-like symptoms, impulsivity, and obesity has been recognized, at least in adults [6,7,8,9] (but see also [10,11]). A potential relationship between disinhibited behavior and experiencing downshift (i.e., consumption of less sweet than expected) has not been documented by the existing literature (in rodents, only consequences of limited access to 32% sucrose and related downshift are examined and negative emotional reactions accompanied by voluntary anxiolytics consumption were recognized [12,13]). Nevertheless, the association between sugar intake and behavioral performances are still debated, particularly concerning that sugar consumption also improves athletic, cognitive, and academic performance and may increase self-control and reduce aggressive behavior, although these effects may be most apparent shortly after sugar has been consumed [14]. More information on the potential relationship between USI-induced body weight (BW) changes, on the one hand, and USI-induced behavioral changes, on the other hand, could help decode this puzzling question and define whether there is any relationship between these two variables. 

Disinhibited personality traits, including impulsivity, sensation seeking, and low fearfulness [15], have been implicated in the development of externalizing behaviors [16,17,18], with reduced hypothalamic-pituitary-adrenocortical (HPA) axis responses to acute stress as the endocrine parameter linked to behavioral disinhibition [4,19,20]. The capacity of highly-palatable foods and added sugar to reduce the HPA axis activity in response to stress has been documented, which can explain an increase in sugar consumption in the conditions of chronic stress [21,22] (reviewed in [23]). However, an adequate stress response is essential for functional adjustments in reaction to ecologically relevant challenges in ordinary life. For example, exposure to novel contexts should increase circulating concentrations of the adrenal stress hormones corticosterone and epinephrine and produce heightened states of arousal [24,25], which is important for the normal physiological reactions required to support behavior [26] and to produce an emotional activation in everyday challenges [27]. Otherwise, as explained above, disinhibited behaviors might appear [4,20], at least in males [19].

A recent study by Beecher et al. [28] showed for the first time, in a mouse model, that long-term consumption of sucrose is related to significant weight gain, hyperactivity, and learning impairments, as well as reduced control to resist the food, while significant changes in emotional behavior have not been detected, suggesting that long-term sugar intake might play a role in the pathogenesis of attention deficits and hyperactivity-related disorders [28]. However, in this study, the exposure of the animals to sucrose solution started in adolescence and implicated single housing, which may contribute to the specificity of the findings.

The present study aimed to examine whether and how the USI and subsequent downshift (having less sweet than expected) influence BW and behavioral inhibition in the paradigms that assess motor/exploratory, anxiety-like, and social dominant behavior in adult male C57BL/6J mice. To provide a model that would incorporate human-like patterns of self-administration in laboratory animals, we opted for a design which allows free choice access and thus applied the two-bottle choice paradigm (2BCP) in male C57BL/6J mice, allowing them unlimited access to the potent sucrose solution (32%) in home cages, without affecting social environment/hierarchy. Indications that added sugar provided by liquid food sources may play a more powerful role in the development of obesity/overweight than added sugar from solid food sources (because it may not result in compensation for energy intake to the same extent as solid added sugar food sources, leading to excess food consumption and resultant weight gain; discussed in [29]) were taken into account. Additional steps, before the highly concentrated sucrose solution is made available to the animals, were introduced to ensure that all groups would start consuming the 32% sucrose at the same time. Considering previous findings on the relationship between sugar intake and blunted HPA axis response to stress, we hypothesize that the animals that consume sucrose solution would have altered response to both non-social and social novelty, expressing fearlessness and social confrontation (i.e., sensation-seeking-like response), with nuances in response due to sucrose downshift. 

## 2. Materials and Methods

### 2.1. Animals

The experiments were performed on adult male C57BL/6J mice (the most common inbred mouse strain employed in biomedical research, summarized in [30]) which were housed in 10 cages (4 animals per cage) to avoid crowding. The animals were from different litters and were placed in their designated cages during puberty, before social dominant behaviors have developed [31], balancing equal proportions of mice from different litters. 

Housing conditions were the same for all animals prior to the experiment, i.e., they had ad libitum access to standard chow and fresh tap water daily, and they were kept within the same room with a 12:12 h light–dark cycle (lights on at 7am) at 23 °C. All animal procedures were in compliance with EU Directive 2010/63/EU and were approved by the Ethical Committee of the Institutes and by the National Ethic Research Committee (323-07-09346/2019-05).

### 2.2. Experimental Procedure

A schematic presentation of the experimental design is given in Figure 1. The entire procedure consisted of a pre-experimental phase, experimental phase 1, and an experimental phase 2. 

In the pre-experimental phase, the animals were handled by the experimenter (who also performed behavioral testing) for approximately 10 min daily (Figure 1; experimental days −10 to −5). Neurological pre-testing of the animals was done, using beam walking test and reflex assessment (negative geotaxis, hind-limb clasping reflex, and righting reflex) to exclude potentially defective subjects (Figure 1; day −4). Two sources of water (adapted syringes, 50 mL) were introduced using 2BCP on day −4 and −3 (Figure 1). Thereafter, on experimental day −2, the animals were subjected to the body weight measurement, and 2% sucrose solution was introduced using 2BCP (Figure 1). On the experimental day −1, 4% sucrose solution was introduced using 2BCP (Figure 1). These steps were done with the purpose to reduce the preference for the source of the liquid located on the right and the effects of neophobia [32]. 

In the experimental phase 1 (Figure 1; experimental day 0–10), the 2BCP was applied with water and 32% sucrose solution (adapted syringes, 50 mL for both; ad libitum access) in home cages for all animals (N = 8 cages; 4 animals per cage) except the control group (N = 2 cages; 4 animals per cage). The position of the bottles was switched daily, and consummatory behavior was measured following a 24 h period to obtain a total of 10 measures. 

The experimental phase 2 started on experimental day 10. Animals that received 32% solutions in the previous phase were randomly assigned into 4 different groups depending on the concentration of sucrose they would receive in this phase (Figure 1; experimental days 10–11). The devaluated solution was introduced to three experimental groups (32%→16%; 32%→8%; 32%→4%; N = 2 cages per group): one experimental group received the unchanged 32% sucrose solution (32%→32%; N = 2 cages), while the control group (N = 2 cages), identical to the previous phase, was not exposed to sucrose. 

Behavioral assessment was conducted after the measurement of consummatory response to the last placement of sucrose solutions was completed. Three tests were conducted. As the exposition to each experimental apparatus may lead to a broad sort of secondary effects, which may impact the results of the following test, the tests were performed in the following order for all groups: Novel Open Arena (NA) → Elevated Plus-Maze (EPM) → Social Confrontation Tube test (SCTT). This particular sequence of testing (from least stressful to most stressful test) was chosen in order to reduce the potential confounding effects of test-related stress [33]. Before behavioral testing, all animals were habituated to the experimental room for 30 min; each mouse had between 1 and 1.5 h of interlude between each test. 

### 2.3. Exploratory/Motor Activity in the Novel Open Arena (NA)

Following a habituation period (30 min in home cages) to the testing room, mice were placed in the Opto-Varimex cages (Version 3.0A, Columbus Inc., Melbourne, OH, USA) and allowed to freely explore the novel environment for 5 min. Each cage (44.2 × 43.2 × 20 cm^3^) was equipped with 15 infrared emitters located on the x and y axes. An equivalent number of receivers were located on the opposite walls of the cage. Data were analyzed using Auto-Track software (Columbus Instruments). The Auto-Track interface collects data from the Opto-Varimex unit every 1/10th of a second and categorizes the activity. Locomotion was defined as a trespass of consecutive infrared beams and vertical activity as the number of infrared beams that were broken by the rearing of the animal. The Auto-Track interface has the ability to detect movements in 16 (4 × 4) equal fictional squares, allowing for the calculation of number of entries and time spent in the central zone (four squares in the middle). 

After termination of the 5-min exploratory period in the novel arena, tested animals were returned to their home cages and boxes were carefully cleaned and deodorized with a 20% ethanol cleaning solution to erase any smells which might interfere with the exploration by the next animal. 

The NA is a standardized test for assessing locomotor activity in rodents, however it is also widely used as a means for assessing anxiety-like behavior, which is under those circumstances operationalized as inversely proportional to time spent in and number of entries into the central zone of the arena. 

### 2.4. Elevated plus Maze (EPM)

The animals were placed in the center area facing the open arm as per the standard EPM protocol [34]. Activity was recorded for 5 min, and the following indicators were subsequently cataloged by two independent researchers via the “blind” procedure: number of open and closed arm entries (entry into the arm is counted from the moment the animal has all four legs in the arm), as well as time spent in the open and closed arms and the central zone. The number of fecal boli left by the animal during the test session was monitored as well.

The EPM is a widely used behavioral test for assessment of anxiety-like behavior in rodents, where anxiety-like behavior is operationalized as inversely proportional to time spent in open arms and number of open arms entries.

### 2.5. Social Confrontation Tube Test (SCTT)

This convenient method allows for measuring aggressive and social dominance traits outside of the home cage in rodents [35]. A plexiglas tube (50 cm in length, with a 3 cm internal diameter; the size of tube used was chosen to ensure that two mice could not pass each other) was firmly attached to the designated surface area in the experimental room. The mouse from the sample was placed near one opening of the tube, while the “opponent” was placed on the other side. The number of wins, when the mouse from the sample had successfully pushed out the opponent or compelled them to withdraw from the tube out of three encounters was the final output and indicator of social dominance. Activity of each animal was recorded by a camera, allowing subsequent analysis of the recorded material by two independent researchers. The mice used as opponents (11 in total) in the SCTT were additional animals of the same age and sex, and similar stature (BW was the same or lower), which were used only for the purpose of this test. They were kept in the same standard housing conditions as the mice from the sample, and were habituated to the presence of the experimenter. The number of opponents was consistent with the protocol because it does not indicate that opponent mice cannot be used multiple times during a test session, as long as the opponent and target mice are of a similar weight and stature. All animals (opponent and target mice) had previous individual experience with the plexiglas—they were encouraged to walk forward through the tube from both ends, 10 times per each side, 2 days before the final testing to minimize the potential effect of learning, as opponents had more overall experience with passing through the tube during testing.

### 2.6. Criteria for Exclusion of Animals

#### 2.6.1. Exclusion Criteria during the Pre-Experimental Phase

Any aggressive behavior detected during pre-experimental monitoring of animals in home cages, presence of injuries, and other signs of violence, barbering, and stress are to be taken as an indicator to exclude such cages from further examination. No cages were excluded from the study based on this criterion.

Deficits in reflexes detected via neurological assessment, as well as an inability to successfully cross the beam during the pretest, should be taken as indicators to exclude such animals from the sample. In this study, no animals were singled-out or excluded based on these criteria. 

Cages in which there was no expenditure of the 2% or 4% sucrose solutions on day −2 and −1 (i.e., where the expenditure of solutions was ≤1mL) should be excluded from the study, as this might be an indicator of gustative deficits. In the current study, no cages were excluded based on this criterion.

#### 2.6.2. Exclusion Criteria during the Experimental Phase 1 and Experimental Phase 2

Seeing as the purpose of the experiment was to evaluate the effect of consuming sucrose solutions on behavioral and consummatory outcomes, any cages in which animals failed to consume the solution should be excluded from further analysis. 

In experimental phase 1, no cages were excluded from the study. In experimental phase 2, two cages were excluded from the study (one that received the 8% solution and one that received 16% solution), as the animals in these cages inexplicably failed to consume any sucrose solution (the expenditure of solutions was 0 mL, indicating a technical problem such as obstruction of syringes). Seeing as the effects of complete omission of sucrose were out of the scope of this study, these cages were not taken into analysis of effects of downshift. 

#### 2.6.3. Exclusion Criteria for Behavioral Assessment

Regarding the results obtained in the NA, any incomplete outputs from the Opto-Varimex Auto-Track system should be taken as an indicator that these outputs should be excluded from the analysis. Similarly, regarding results obtained in the EPM and SCTT, all video recordings that are of low quality so that quantification would be problematic should be excluded from the analysis.

No results obtained during behavioral assessment in this study were excluded, based on these criteria. 

### 2.7. Statistical Analysis

Statistical analysis was performed using Statistica 6.0 software (StatSoft Inc., Tulsa, OK, USA). Graphical and tabular presentation of results is given. Qualitative data were expressed as percentages, while quantitative data were expressed as means ± standard deviation (SD), with individual data plots along the column bars. Statistical significance in all statistical comparisons was accepted with *p* ≤ 0.05. 

The analysis of changes in consummatory activity (per cage) during experimental phase 1 was performed using Friedman ANOVA, followed by Wilcoxon matched pairs test, as some datasets did not have a normal distribution. Within-subject design was used since it allows each subject to be their own “control”. Such a design has the advantage of controlling extraneous participant variables, making it easier to detect the relationship between independent and dependent variables than with between-subject design. 

Seeing that after downshift the number of samples in the two experimental groups was reduced due to the exclusion of animals (according to the exclusion criteria), thus leading to unequal sample size per group, and that some datasets did not have a normal distribution, the results obtained in behavioral tests related to experimental phase 2 were analyzed using nonparametric statistics. Comparisons with the control group were done using Mann–Whitney U test. The factor of sucrose downshift on examined behavioral parameters was analyzed using Kruskal–Wallis ANOVA followed by Mann–Whitney U test. 

Multiple testing corrections were performed using the Benjamini–Hochberg procedure to control the false discovery rate, and only those *p* values that passed the criteria posted by the procedure were considered significant. Controlling false discovery rate has been recognized as appropriate when the goal is to identify candidate effects from a large set and recognize groups that can be tested more rigorously in follow-up confirmatory experiments [36].

To get additional information on the relation between social dominance and the measures of anxiety and exploratory activity, we used Kendall’s tau correlation to measure the strength of dependence between winning points in a social confrontation tube test and the parameters of anxiety and general activity obtained in the EPM and NA. This was done in accordance with the nature of obtained data, as Kendall’s tau is appropriate when working with small samples and multiple values with the same score [37]. Tabular presentation of obtained findings is available in the Results section.

## 3. Results

### 3.1. Food and Liquid Intake Due to Unlimited Access to 32% Sucrose Solution in the Two-Bottle Choice Paradigm

Parameters of consummatory behavior of adult C57BL/6J male mice are presented in Figure 2. Total volume of food and water intake two days before introducing the two-bottle choice paradigm is available as well, and these values were used as internal control data. Please note that only cages with animals that passed all exclusion criteria (and animals contributed to the behavioral results) were taken into consideration. Therefore, the number of cages is N = 6 (as two cages were excluded; for details please see Section 2.6.2. Exclusion criteria during the experimental phase 1 and experimental phase 2).

The amount of consumed food during ad libitum access to 32% sucrose solution was lower compared to the internal control data (Figure 2A; *p* = 0.03, Z = 2.20, Wilcoxon matched pairs test), but after the application of the Benjamini–Hochberg procedure these effects became insignificant. Additionally, the amount of consumed food during this time did not vary significantly (Figure 2A; χ^2^(N = 6, df = 9) = 10.11, *p* < 0.34). 

Compared to the internal control values (Wilcoxon matched pairs test), total volume intake during unlimited 32% sucrose solution access was not seen as significant after the application of the Benjamini–Hochberg procedure, as well as the total volume intake during the period of 32% sucrose solution access.

The Friedman ANOVA revealed significant variations in the volume of sucrose solution consumed across days in the two-bottle choice paradigm (Figure 2C; χ^2^(N = 6, df = 9) = 17.61, *p* < 0.04), but post-hoc comparisons, performed using Wilcoxon matched pairs test and subsequently the Benjamini–Hochberg procedure, remained insignificant. 

There were no significant variations in the amount of water consumed across 10 days during the two-bottle choice paradigm (Figure 2D; χ^2^(N = 6, df = 9) = 12.09, *p* < 0.21). Even when values representing total fluid intake were used as internal control values for water intake (the total volume intake for the control measures also reflects water intake as for this measure the animals had no experience with sucrose, i.e., water was the only available liquid), within-group comparisons (performed using Wilcoxon matched pairs test) followed by BHP remained insignificant (in Figure 2, water intake in control conditions is presented as total liquid intake for practical reasons, due to the scale range on the *Y*-axis).

### 3.2. Body Weight (BW)

Data related to the BW of mice during the 13-day period (2 days before and 10 days during 32% sucrose consumption and 1 day of the downshift) are shown in Figure 3, showing that the presence of sucrose did not affect BW of the animals, i.e., did not induce overweight. 

### 3.3. Behavior of the Animals in the Novel Open Arena (NA)

Data related to animal activities in the NA are shown in Figure 4. Locomotor activity (Figure 4A) was highly similar in the control and sucrose exposed groups. Sucrose downshift appeared as an important factor for locomotor activity (H (3, N = 24) = 7.76, *p* = 0.05), but post-hoc analysis followed by the Benjamini–Hochberg procedure did not reveal significant differences. Considering vertical/rearing activity (Figure 4B) and the number of entries in the central zone of the arena (Figure 4C), there were no significant differences between the control and sucrose exposed groups (*p* ≥ 0.14) and the downshift did not significantly influence these parameters (H (3, N = 24) = 4.84, *p* = 0.18 and H (3, N = 24) = 4.78, *p* = 0.19, respectively). 

The time spent in the central zone of the novel open arena (Figure 4D) was significantly increased in the group that had access to 32% sucrose for the entire time compared to the control group (Figure 4D; * *p* < 0.01, Z = −2.78, U test followed by the Benjamini–Hochberg procedure), while the significance was not detected in the downshifted groups (*p* ≥ 0.13); the downshift per se was not a significant factor for the time spent in the central zone of the novel open arena (H (3, N = 24) = 4.33, *p* = 0.23).

### 3.4. Behavior of the Animals in the Elevated plus Maze (EPM)

Data related to animal activity in the EPM are shown in Figure 5. 

The number of entries into the closed arms (Figure 5A) was control-like in all sucrose-exposed groups. Sucrose downshift appeared as important factor for the number of entries in the closed arms (H (3, N = 24) = 9.29, *p* = 0.03); post-hoc analysis revealed that 32→8 downshifted group had significantly more entries in the closed arms than the group that had access to 32% sucrose for the entire duration of the two-bottle choice period (Figure 5A; & *p* < 0.01, Z = −2.72, U test followed by the Benjamini–Hochberg procedure). 

The number of entries into the open arms (Figure 5B) was significantly increased in all sucrose-exposed groups compared to the control group (Figure 5B; * *p* ≤ 0.03, −3.36 ≤ Z ≤ −2.72, U test followed by Benjamini–Hochberg procedure) and the downshift was not an important factor for this parameter (H (3, N = 24) = 3.79, *p* = 0.28). 

The total number of arm entries (Figure 5C) was significantly increased in all sucrose-exposed groups compared to the control group (Figure 5C; * *p* ≤ 0.01, −2.72 ≤ Z ≤ −2.21, U test followed by Benjamini–Hochberg procedure). The downshift appeared as an important factor for this parameter (H (3, N = 24) = 8.62, *p* = 0.03) and, as post-hoc analysis revealed, 32→8 downshifted group had significantly higher total number of entries than the group that had access to the 32% sucrose for the entire time (Figure 5C; and *p* < 0.02, Z = 2.38) as well as than 32→4 downshifted group (Figure 5C; # *p* = 0.01, Z = 2.55, U test followed by Benjamini–Hochberg procedure). 

When the number of entries into open arms was corrected for the total number of entries (Figure 5D), all sucrose exposed groups still had increased open arms activity compared to the control groups (Figure 5D, * *p* < 0.02, −3.26 ≤ Z ≤ −2.38, U test followed by Benjamini–Hochberg procedure). Sucrose downshift did not exert significant influence on this parameter (H (3, N = 24) = 4.57, *p* = 0.21). 

The time spent in the closed arms (Figure 5E) was significantly decreased only in 32→8 and 32→4 downshifted groups compared to the control (Figure 5E; * *p* < 0.01, 2.72 < Z < 2.78, U test followed by Benjamini–Hochberg procedure) and the downshift did not significantly influence this parameter (H (3, N = 24) = 5.17, *p* = 0.16). 

All sucrose-exposed groups spent more time in the open arms than the control group (Figure 5F; * *p* < 0.03, −3.15 ≤ Z ≤ −2.21, U test followed by Benjamini–Hochberg procedure), and the downshift did not significantly influence this parameter (H (3, N = 24) = 4.12, *p* = 0.25). 

The time spent in the center (Figure 5G) was control-like in all sucrose exposed groups and the downshift was not an important factor for this parameter (H (3, N = 24) = 5.81, *p* = 0.12). 

The number of fecal boli produced in the EPM was highly similar between the control group and sucrose-exposed groups (Figure 5H); the downshift did not significantly influence this parameter (H (3, N = 24) = 6.56, *p* = 0.09). 

### 3.5. Behavior of the Animals in the Tube Test

Data related to animal behavior in the tube test are shown in Figure 6. The winning points for each group (Figure 6A), calculated by counting the number of wins (0, 1, 2, or 3) in a match with the same “opponent” mouse, did not show statistical differences between the control and sucrose exposed groups (*p* ≥ 0.14), and the downshift did not appear as a significant factor for the number of wins in the tube (H (3, N = 24) = 1.14, *p* = 0.77). That is, percentage of wins per match (Figure 6B) for the control group was 13%, while sucrose exposed groups had 25% (the group that had free access to 32% sucrose solution for the entire time i.e., no downshift), 42% (32%→16% downshift), 33% (32%→8% downshift), and 42% (32%→4% downshift).

To get better insights into the relationship between the behavior of the animals in the tube test and the measures of anxiety and exploratory activity, we performed Kendall’s tau correlation to measure the strength of dependence between winning points in a social confrontation tube test and the parameters of anxiety and general activity obtained in the EPM and NA. Obtained findings are given in Table 1. For the control group, a positive correlation was obtained for the time spent in the closed arms of the EPM (Table 1; Kendall’s τ = 0.577, *p* = 0.046) and the number of fecal boli in the EPM (Table 1; Kendall’s τ = 0.605, *p* = 0.036), while negative correlation was obtained for the number of entries into open arms of the EPM (Table 1; Kendall’s τ = −0.591, *p* = 0.041) and the number of entries into the central zone of the NA (Table 1; Kendall’s τ = −0.653, *p* = 0.024), overall suggesting an anxious behavioral phenotype of control winners in the tube test. For the group that had ad libitum access to 32% sucrose solution, a positive correlation was obtained for the time spent in the open arms of the EPM (Table 1; Kendall’s τ = 0.592, *p* = 0.040), while negative correlation was obtained for the time spent in the closed arms of the EPM (Table 1; Kendall’s τ = −0.592, *p* = 0.040), overall suggesting that wins belong to less anxious animals of this experimental group. For other experimental groups, no relations between winning and behavioral parameters of the EPM and NA were observed (Table 1). 

## 4. Discussion

Increased sugar consumption becomes an important public health concern and there is a need to properly understand consequences. The present study shows, in the mouse model, that USI does not influence animals’ BW but produces anxiolytic-like effects and, depending on the characteristics of the novel environment, hyperactivity. After the downshift in the concentration of the sucrose solution, behavioral effects strongly depend on the characteristics of the novel environment and are evident in the EPM but not in the NA paradigm. USI does not influence successes/wins in an episode of social conflict, but it seems to alter the emotions which drive the winning, favoring a less anxious behavioral phenotype, which was not observed in the downshifted groups. Observed findings suggest that USI promotes sensation-seeking and motivates dominance without changing BW, while a blunted emotional base of social dominance might be an early mark of the downshift. Nevertheless, the generated hypothesis about the relationship between USI and externalizing behavior in the mouse model remains to be confirmed in a larger, confirmatory study, with a larger number of experimental animals per group and different lengths of exposure to USI conditions. Deepening this topic may help to gain better insight into the behavioral autograph of a sedentary lifestyle and ad libitum sugar consumption, which are becoming an important public health problem.

In this study we attempted to characterize the model for the assessment of consequences of ad libitum access and voluntary intake of sugar, as well as consequences of getting less sweetness than expected, using a free choice paradigm. We observed that the consumption of ad libitum available 32% sucrose solution per cage, in 2BCP, was balanced during the 11-day period (in accordance with the 10 obtained measurements). The literature describes difficulties in forcing animals to drink a high percentage of sucrose solution and the reduction of the amount of available food is advised to motivate them to do so [38]. We believe that in our study, initial two-day access to 2% and 4% sucrose solutions, as well as ensuring that animals remain in the social groups (as opposed to being isolated), contributed to balanced 32% sucrose consumption during the period of interest (although in this way individual measures were lost). Our study did not reveal significant differences in the consummatory activity of experimental animals in the presence of 32% sucrose solution, and the small sample size likely contributed to this finding. However, a trend towards lower food intake in the presence of sucrose than in the presence of tap water (i.e., under standard conditions) was observed, and as such remains to be verified in further studies with a larger sample. This phenomenon has previously been described in the literature dealing with daily intermittent sugar intake [3] as a compensatory response to additional calories derived from sugar, resulting in a normal body weight. It should be noted that food and water intake measured in our study per cage with 4 mice (16 g and 26.8 mL, on average), in standard conditions, is consistent with data on the amount of food consumed and water drunk per mouse, in C57BL/6J mice (4 g and 6–7 mL per mouse [39]). Previous findings also showed that socially housed C57BL/6J mice do not show weight gain when observing the period around postnatal days 100 to 126 [40] (see also [41]). 

Motor activity, along with free-choice responses to novel environments that differ in complexity and aversiveness, have been used to assess anxiety and sensation-seeking in rodents [42] as the response to novelty is the basis of the definition of sensation-seeking [43]. Our study reveals that USI does not produce an increase in locomotor and vertical activity of the animals (32→32 experimental group) in the NA, however the number of entries and time spent in the central zone of the arena were increased compared to the control group. An increase in central activity or in time spent in the central part of the arena, without modification of total locomotion and of vertical exploration, can be interpreted as an anxiolytic-like effect of unlimited sucrose intake without general stimulation [44]. Concerning motor activity of the control group in the NA, it should be noted that the locomotor and vertical/rearing activity values obtained for 5 min registration period in our study are almost the same as those reported in the study by Trullas and Skolnick [45] using C57BL/6J mice and the same equipment (Opto-Varimex computerized activity monitor, Columbus Instruments, Columbus, OH). Importantly, such central activity was not observed in the downshifted groups of mice, suggesting that they respond differentially to the NA (an inescapable novel environment) than the 32→32 group and are instead more similar in this sense to the control group. This might mean that approach-avoidance motivation was balanced in downshifted groups, so their physiological arousal was at the control-like level and the central zone of the NA was not attractive to them. Nevertheless, the interpretation of changes in the systems controlling approach, avoidance, and approach-avoidance conflict (reviewed in [46]) overcomes the purpose of this study. 

It is a known fact that different testing environments could produce different behavioral profiles [33]. The present study reveals that USI- and the downshift-induced behavioral response to environmental novelty was more emphasized in the EPM than in the NA, as all sucrose-exposed groups showed an increase in open arm entries and time spent in the open arms, i.e., less anxiety-like behavior than the control group [47]. Moreover, all sucrose-exposed groups showed increased motor activity in the EPM, which was evident through an increase in the number of total arm entries [47]. The time spent in the central zone, an indicator of a decision-making period [45], was not changed. Considering that seeking emotional stimulation, along with active exploration of an environment that is typically considered anxiogenic and stressful, are characteristics of sensation-seeking in experimental animals [42,48], our findings strongly indicate that USI promotes sensation-seeking, with the characteristics of environmental novelty as highly important for the manifestation of sensation-seeking after the downshift. Low arousal (under-arousal) may implicate stimulation-seeking to increase arousal levels near physiological optimum [49,50]. It is known that arousal differs with the intensity of experienced stimulation [51], and potential differences in the experienced stimulation of the downshifted groups by NA and EPM can explain the absence of sensation-seeking-like behavior in the NA and its presence in the EPM. 

In the present study we attempted to assess the potential impact of USI and the downshift on social behavior within the context of the group. Using SCTT, we observed that neither USI not the downshift influenced successes/wins in the test. However, correlation analysis revealed that in the control group of C57BL/6J mice dominant ones exhibited anxiety-like behavioral profile (which is in line with the recent findings by Larrieu et al. [52]), while in the group that had unlimited access to 32% sucrose solution for the entire duration of the experiment, the winners were mice with anxiolytic-like behavioral phenotype. The downshifted groups showed a blunted emotional base of social dominance. These findings indicate that USI alters the emotions that drive the winning in a social confrontation, favoring a less anxious behavioral phenotype, while blunted emotional base of social dominance represent an early mark of the downshift. These findings also suggest that generally similar behavioral phenotypes between 32→32, 32→16, 32→8, and 32→4 groups actually may have different backgrounds. From the methodological point of view, these data agree that the SCTT may reflect other behavior that is not necessarily associated with dominance per se [53,54], drawing attention to the potential but not obligatory impact of anxiety parameters on winning the SCTT.

An important variable to consider is the duration of sucrose exposure. Defining the length of the experimental treatment largely depends on what the study wants to model (specific conditions in one segment of a certain life stage or in the whole life stage or in the whole life span). In the present study, we attempted to model several-years-long unlimited sugar consumption during adulthood. In order to adequately set a period of the exposure, we considered that one day in the life of a human and a mouse do not mean the same, as mice have a diminutive lifespan compared to humans and, during the adult phase, and 2.60 mice days are equivalent to one human year (for a detailed explanation see [55]). Accordingly, 10 days in the life of an adult mouse should be seen as about 3.8 years of the life of an adult human. 

Although this study has notable advantages, limitations should be noted as well. Females have not been included in the study because of the potential variability related to the estrous cycle. Therefore, findings described in this paper should be considered gender-specific unless confirmed in female mice as well. The level of stress hormones was not analyzed; however, these analyses were consciously omitted to avoid the potential effect of stress caused by blood or urine sampling on the behavioral parameters, which were a priority for model consideration and evaluation. Consummatory behavior was not measured in individual (consumption) chambers, thus information related to individual consummatory behavior was lost; this was purposely discarded, as obtaining these measures would be done at the expense of maintaining valid experimental conditions (i.e., avoiding potential effects of social isolation). Small sample size probably contributed to the absence of statistically significant changes in consummatory behavior. Considering sample size, we are aware that a too small sample size can miss the real effect in an experiment, and that a sample size that is larger than necessary will lead to wasting resources and ethical issues on the used animals [56,57]. In this view, we have to note that having 8 animals per group for each of the 5 experimental groups, as we initially started, is quite an acceptable number for our exploratory study and interest in finding any level of difference between groups. Considering resource equation method, this number should be 5 per group (to satisfy the condition that the degree of freedom is not higher than 20 [56,57]), but bearing in mind specific methodological approach used in the present study, as well as our previous experience with the used tests, we opted for 8 animals per group (so at the beginning of the experiment each of 5 experimental groups had 2 cages with 4 animals in each). Although the number of animals per group in two groups was reduced, i.e., the animals were excluded according to the exclusion criteria, we are of the opinion that the obtained results should be considered for publication because the methodological approach itself was not compromised, and therefore there is no reason to discriminate/consider irrelevant the results obtained using other animals. Nevertheless, small sample size is a limitation of the study, and confirmation of the obtained data is undoubtedly needed.

## 5. Conclusions

This study contributes to understanding how unlimited sugar intake affects choice behavior. In literature, this manner of sugar intake has been mainly associated with metabolic consequences. Obtained findings indicate that USI per se does not necessarily produce changes/gains in BW. Instead, they accentuate a central effect of unlimited sugar intake, as well as those of experiencing devaluation, with the increased propensity for sensation seeking as the most important observed behavioral outcome. We hope that the new model will attract the attention of the scientific community, so that the findings of this study will be evaluated in a larger, confirmatory study, and further comprehensively characterized, taking into account gender, age, and other psychobiological circumstances.

## Figures and Tables

**Figure 1 brainsci-12-01332-f001:**
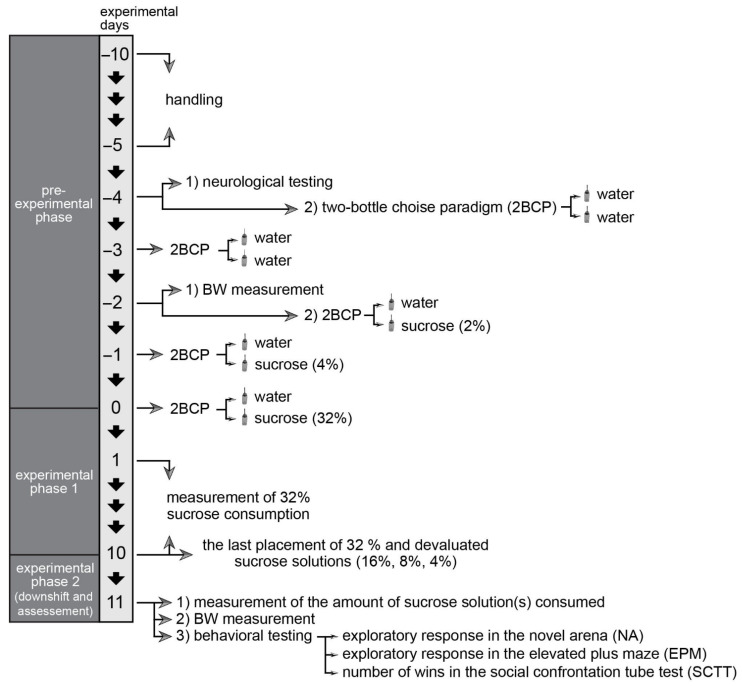
Schematic presentation of the experimental design. A day-by-day overview of the procedure is displayed with all undertaken steps and focal points of the experiment.

**Figure 2 brainsci-12-01332-f002:**
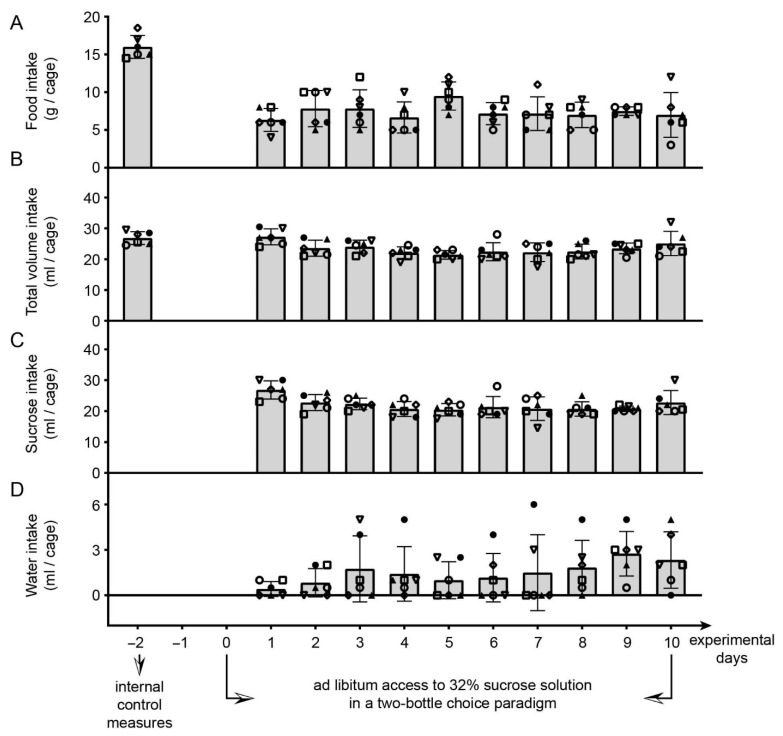
Consummatory activity of adult male C57BL/6J mice associated with ad libitum access to 32% sucrose solution in a two-bottle choice paradigm. Obtained results ((**A**) Food intake; (**B**) Total volume intake; (**C**) Sucrose intake; (**D**) Water intake) are expressed as intake(s) per cage. Note that the total volume intake for the control measures also reflects water intake as for this measure the animals had no experience with sucrose (i.e., water was the only available liquid). Detailed statistical analysis (within-subject design, Wilcoxon matched pairs test) followed by Benjamini–Hochberg procedure did not reveal significant differences between measurements (including the condition when values representing total fluid intake were used as control values for water intake, although not given as a part of panel (**D**)).

**Figure 3 brainsci-12-01332-f003:**
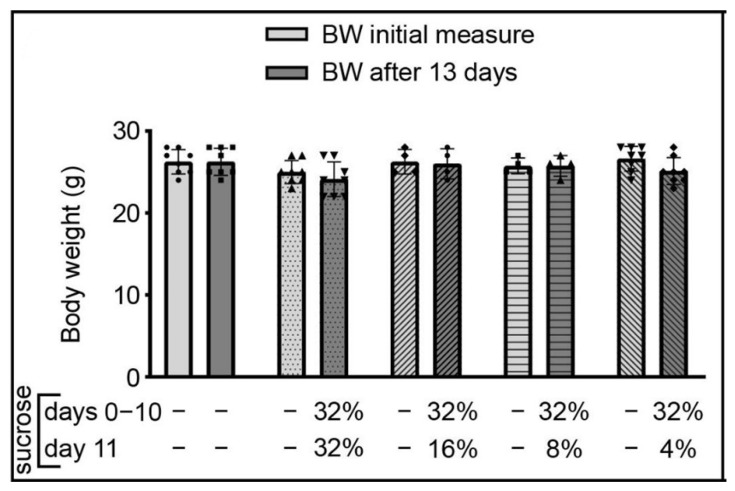
Changes in body weight of adult male C57BL/6J mice associated with ad libitum access to sucrose solution(s). The data are represented as mean ± SD, with individual data plots along column bars (8 animals per group in all except in 32%→16% and 32%→8% downshifted groups which have 4 animals per group). The initial BW measurement was performed 2 days before the introduction of 32% sucrose solution, which was made available to animals on experimental day 0 up to experimental day 10. On experimental day 10, part of the animals/cages received solutions less sweet than expected (they were offered 16%, 8%, or 4% sucrose solutions) for the next 24 h (experimental day 11), when body weight was measured again.

**Figure 4 brainsci-12-01332-f004:**
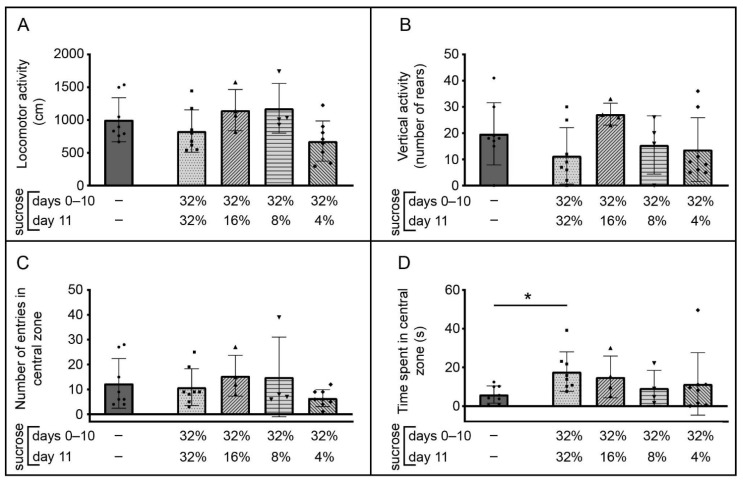
Changes in behavior of adult male C57BL/6J mice in the inescapable novel arena associated with ad libitum access to sucrose solution(s). The data are represented as mean ± SD, with individual data plots along column bars (8 animals per group in all except in 32%→16% and 32%→8% downshifted groups, which have 4 animals per group). Locomotor activity (**A**), vertical activity (**B**), number of entries in the central zone of the arena (**C**), and time spent in the central zone of the arena (**D**) were recorded for 5 min, starting immediately after placing the animal in the monitoring arena, with automated collection of behavioral parameters. * *p* ≤ 0.05 vs. the control group, which had no experience with sucrose, U test followed by Benjamini–Hochberg procedure. The influence of the downshift was not assessed as significant.

**Figure 5 brainsci-12-01332-f005:**
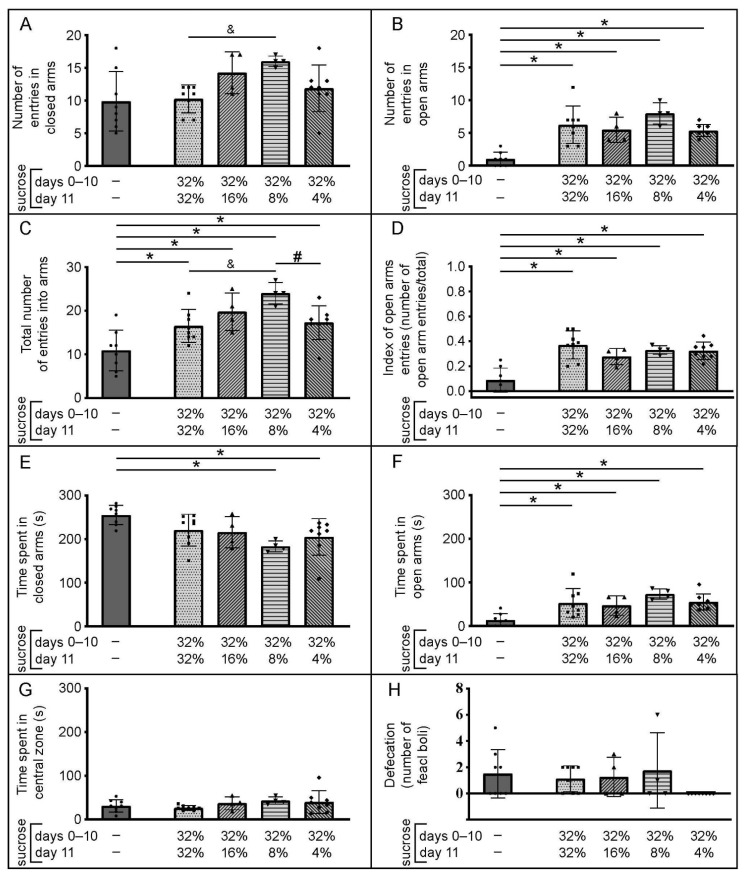
Changes in behavior of adult male C57BL/6J mice in the elevated plus maze associated with ad libitum access to sucrose solution(s). Panels A–H represent different parameters scored in the elevated plus maze test or calculated using scored ones ((**A**) Number of entries in closed arms; (**B**) Number of entries in open arms; (**C**) Total number of entries into arms; (**D**) Index of open arms entries; (**E**) Time spent in closed arms; (**F**) Time spent in open arms; (**G**) Time spent in central zone; (**H**) Defecation). The data are represented as mean ± SD, with individual data plots along column bars (8 animals per group in all except in 32%→16% and 32%→8% downshifted groups, which have 4 animals per group). The behavior of the animals was monitored/recorded during the first 5 min after the placement in the center of the maze and the results were obtained by subsequent analysis of the recorded material. * *p* ≤ 0.05 vs. the control group, which had no experience with sucrose, U test followed by Benjamini–Hochberg procedure. The influence of the downshift: # *p* ≤ 0.05 vs. 32%→4% downshifted group, U test followed by Benjamini–Hochberg procedure; & *p* ≤ 0.05 vs. 32%→32% group, U test followed by Benjamini–Hochberg procedure.

**Figure 6 brainsci-12-01332-f006:**
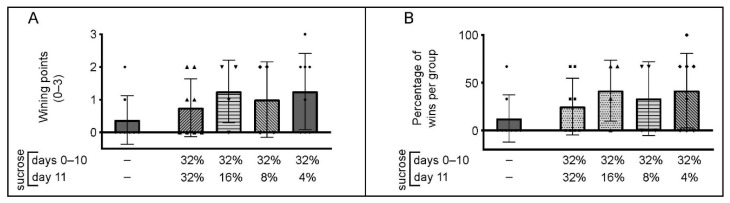
Changes in behavior of adult male C57BL/6J mice in the social confrontation tube test associated with ad libitum access to sucrose solution(s). The data are represented as mean ± SD, with individual data plots along column bars (8 animals per group in all except in 32%→16 % and 32%→8% downshifted groups which have 4 animals per group). Panel (**A**) represents the number of wins (out of three encounters). Panel (**B**) represents the same data expressed as percentage of wins per group. No significant differences between groups are observed.

**Table 1 brainsci-12-01332-t001:** Kendall’s tau correlation used to measure the strength of dependence between winning points in a. social confrontation tube test and the parameters of anxiety and general activity obtained in the EPM and NA.

Tube Wins and Parameters of Behavior in the EPM		Control Group	32%	32% to 16% Downshift	32% to 8% Downshift	32% to 4% Downshift
Tube wins and entries in open arms	Valid N	8	8	4	4	8
	Kendall’s Tau	−0.591	0.228	0.400	0.000	0.186
	Z	−2.048	0.791	0.815	0.000	0.645
	*p*-value	0.041	0.429	0.415	1.000	0.519
Tube wins and entries in closed arms	Valid N	8	8	4	4	8
	Kendall’s Tau	−0.052	0.093	−0.400	0.000	0.213
	Z	−0.182	0.323	−0.815	0.000	0.739
	*p*-value	0.856	0.747	0.415	1.000	0.460
Tube wins and time in open arms	Valid N	8	8	4	4	8
	Kendall’s Tau	−0.555	0.592	0.000	0.000	−0.287
	Z	−1.922	2.049	0.000	0.000	−0.995
	*p*-value	0.055	0.040	1.000	1.000	0.320
Tube wins and time in closed arms	Valid N	8	8	4	4	8
	Kendall’s Tau	0.577	−0.592	0.183	0.000	0.205
	Z	1.997	−2.049	0.372	0.000	0.711
	*p*-value	0.046	0.040	0.710	1.000	0.477
Tube wins and number of fecal boli	Valid N	8	8	4	4	8
	Kendall’s Tau	0.605	−0.410	0.000	0.894	∕
	Z	2.097	−1.422	0.000	1.823	∕
	*p*-value	0.036	0.155	1.000	0.068	∕
Tube wins and parameters of behavior in the arena (NA)		Control group	32%	32% to 16% downshift	32% to 8% downshift	32% to 4% downshift
Tube wins and locomotor/ambulatory activity	Valid N	8	8	4	4	8
	Kendall’s Tau	−0.472	0.169	0.183	0.000	−0.403
	Z	−1.634	0.586	0.372	0.000	−1.396
	*p*-value	0.102	0.558	0.710	1.000	0.163
Tube wins and vertical/rearing activity	Valid N	8	8	4	4	8
	Kendall’s Tau	−0.107	0.000	−0.548	−0.816	−0.205
	Z	−0.370	0.000	−1.116	−1.664	−0.711
	*p*-value	0.712	1.000	0.264	0.096	0.477
Tube wins and time spent in the central zone	Valid N	8	8	4	4	8
	Kendall’s Tau	−0.472	−0.085	0.548	0.000	−0.242
	Z	−1.634	−0.293	1.116	0.000	−0.837
	*p*-value	0.102	0.770	0.264	1.000	0.402
Tube wins and number of entries in the central zone	Valid N	8	8	4	4	8
	Kendall’s Tau	−0.653	−0.089	−0.183	0.408	−0.502
	Z	−2.261	−0.310	−0.372	0.832	−1.738
	*p*-value	0.024	0.757	0.710	0.405	0.082

## Data Availability

Data supporting the reported results are contained within the article and are available on request from the corresponding author.

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
