# Peer review of "The Influence of Unlimited Sucrose Intake on Body Weight and Behavior—Findings from a Mouse Model"

_brainsci, 2022, doi:10.3390/brainsci12101332_

Round 1

Reviewer 1 Report

This is potentially interesting study examining the effect of sucrose exposure on body weight and behavior. The authors essentially hardly find any effects. I suspect this is because of two reasons: very low sample size and short exposure to sucrose. For the data to support any conclusions, the study needs to be replicated in house at least once or twice and the appropriate statistical comparisons must be applied. I would further recommend running the same tests but after 3 or 6 months of sucrose exposure. Comparisons across the 10 day and a long-term exposure would be very high impact in my opinion. I do understand that such long experiments may be beyond the scope of this work, but in house replication of the current experiments and increasing the sample size should be perfectly achievable.

Major issues

1) This study is under sampled. It appears from the Methods that the authors ran a study of 10 cages, 4 mice in each, in a single large experiment. After excluding 2 cages for, in my opinion, perfectly valid criteria, they were left with 8 cages, and thus only 4 animals in some groups. Having done similar behavioral experiments myself, a critical step prior to publication is self-replication to make sure the effects are not due to a single batch. Besides, 4-8 mice in a behavioral study where sample sizes of 20-30 animals, spread across 3-4 replications are typically required to call results reliable, is problematic to say the least. I strongly encourage the authors to repeat the study in at least one or two times.

2) Statistical comparisons:

Figure 2: the authors compare each day’s results to either the day before introducing 32% sucrose or the first day of sucrose exposure. They call this a “within-subject design”, which is fine, but they do not account for the multiple comparison nature of the statistics. When comparing the same (day -2) data to a multitude of other datasets (days 1-10), the statistics must account for multiple comparisons post hoc. Wilcoxon Each Pair test may be more appropriate or simply correcting the Wilcoxon Matched Pair for multiple comparisons (fair warning: with 10 comparisons, statistical differences at the one sigma level can only be reported at p values below 0.005). I suspect very few of the Figure 2 results will show significant differences from the “internal control” when the statistics are calculated correctly. I find it odd that the authors chose not to show the data from the water only mice in Figure 2, as a true control group, it would be important to see how they faired over the time of the experiment.

Figure 4 suffers from the same issue: the authors seem to liberally utilize the Mann-Whitney U test (which is meant to do a one-time comparison of two populations) to compare 5 groups against the control and each other (20 different permutations?). This statistical method is woefully inadequate for this task. I agree that there may be some differences and trends here, but the statistics must be adjusted for the multiple comparisons used here.

3) Claiming that downshift may be an important factor for locomotion is a bit misleading: 16% and 8% downshift is no different from control, only different from 4% downshift? How is this meaningful is unclear. I recommend that the authors clarify that there is no difference to control, I am not convinced the measured difference is not simply due to under sampling.

4) One cannot help but wonder if the lack of effects across the paper are due to the relatively short exposure to sucrose. In humans, 10 days of sugar consumption is not likely to have any measurable effect either. A more appropriate timescale for this experiment would be at least a couple of months on high sugar diet.

5) The authors make a lot of claims about how their animal model (both the sucrose consumption and the behavioral paradigms) is related to various human conditions. I recommend dialing this rhetoric back a bit as mouse behavior in an open field or in an EPM is difficult to directly relate to human psychology.

Minor problems

1) References in the Introduction are rather scarce. There are some big claims made in the Introduction related to a variety of things ranging from obesity, sugar intake and ADHD, attention, and other behaviors, all with a single citation. Since these are meant to motivate the study, extensive citations are needed to showcase the robustness of the authors’ claims.

2) Line 115-116: take out “the most common 115 inbred mouse strain employed in biomedical research”, this appears to be the authors’ opinion, not substantiated by anything.

3) Figure 2: data missing. I understand that 2 cages were taken out of the study due to not consuming sucrose during the downshift but the way the Methods are written suggests that they were only taken out from downshift and after. Yet, I only see 6 datapoints in figure 2 instead of 8. Why are there data points missing?

Author Response

Reviewer 1

Reviewer 1, comment 1. This is potentially interesting study examining the effect of sucrose exposure on body weight and behavior. The authors essentially hardly find any effects. I suspect this is because of two reasons: very low sample size and short exposure to sucrose. For the data to support any conclusions, the study needs to be replicated in house at least once or twice and the appropriate statistical comparisons must be applied. I would further recommend running the same tests but after 3 or 6 months of sucrose exposure. Comparisons across the 10 day and a long-term exposure would be very high impact in my opinion. I do understand that such long experiments may be beyond the scope of this work, but in house replication of the current experiments and increasing the sample size should be perfectly achievable.

Response to Reviewer 1, comment 1. Thanks for the comment, the answers are contained in the responses to your comments 2 and 6. Also, we have expanded the discussion (all additions are marked in red) according to your comments considering that they concern very important but sometimes neglected issues in experimental work regarding the type of study and length of treatment.

Major issues

Reviewer 1, comment 2. This study is under sampled. It appears from the Methods that the authors ran a study of 10 cages, 4 mice in each, in a single large experiment. After excluding 2 cages for, in my opinion, perfectly valid criteria, they were left with 8 cages, and thus only 4 animals in some groups. Having done similar behavioral experiments myself, a critical step prior to publication is self-replication to make sure the effects are not due to a single batch. Besides, 4-8 mice in a behavioral study where sample sizes of 20-30 animals, spread across 3-4 replications are typically required to call results reliable, is problematic to say the least. I strongly encourage the authors to repeat the study in at least one or two times.

Response to Reviewer 1, comment 2. Thank you for the comment.  The following answer also covers part of the question posed in comment 1. Please note that this response is incorporated in the discussion section.

Considering sample size, we are aware that a too small sample size can miss the real effect in an experiment, as well as that a sample size that is larger than necessary will lead to wasting resources and ethical issues on the used animals [56, 57]. In this view, we have to note that having 8 animals per group for each of the 5 experimental groups, as we initially started, is quite an acceptable number for our exploratory study and interest in finding any level of difference between groups. Considering resource equation method, this number should be 5 per group (to satisfy the condition that the degree of freedom is not higher than 20 [56, 57]), but bearing in mind specific methodological approach used in the present study, as well as our previous experience with the used tests, we opted for 8 animals per group (so at the beginning of the experiment each of 5 experimental groups had 2 cages with 4 animals in each). Although the number of animals per group in two groups was reduced, i.e. the animals were excluded according to the exclusion criteria, we are of the opinion that the obtained results should be considered for publication because the methodological approach itself was not compromised and therefore there is no reason to discriminate/consider irrelevant the results obtained using other animals. Nevertheless, small sample size is a limitation of the study and confirmation of the obtained data is undoubtedly needed.

  1. Charan, J., & Kantharia, N. D. (2013). How to calculate sample size in animal studies?. Journal of pharmacology & pharmacotherapeutics4(4), 303–306. https://doi.org/10.4103/0976-500X.119726
  2. Arifin, W. N., & Zahiruddin, W. M. (2017). Sample Size Calculation in Animal Studies Using Resource Equation Approach. The Malaysian journal of medical sciences : MJMS24(5), 101–105. https://doi.org/10.21315/mjms2017.24.5.11

Also, please see the first paragraph in the discussion section, which after a brief presentation of the obtained results emphasizes:   

Nevertheless, the generated hypothesis about the relationship between USI ​​and externalizing behavior in the mouse model remains to be confirmed in a larger, confirmatory study, with a larger number of experimental animals per group and different lengths of exposure to USI conditions. Deepening this topic may help to gain better insight into the behavioral autograph of a sedentary lifestyle and ad libitum sugar consumption, which are becoming an important public health problem.

2) Statistical comparisons:

Reviewer 1, comment 3. The authors compare each day’s results to either the day before introducing 32% sucrose or the first day of sucrose exposure. They call this a “within-subject design”, which is fine, but they do not account for the multiple comparison nature of the statistics. When comparing the same (day -2) data to a multitude of other datasets (days 1-10), the statistics must account for multiple comparisons post hoc. Wilcoxon Each Pair test may be more appropriate or simply correcting the Wilcoxon Matched Pair for multiple comparisons (fair warning: with 10 comparisons, statistical differences at the one sigma level can only be reported at p values below 0.005). I suspect very few of the Figure 2 results will show significant differences from the “internal control” when the statistics are calculated correctly. I find it odd that the authors chose not to show the data from the water only mice in Figure 2, as a true control group, it would be important to see how they faired over the time of the experiment.

Response to Reviewer 1, comment 3. Thank you for the comment, it is quite correct. Given that multiple comparisons are present throughout the study, the outcomes of all statistical comparisons were subjected to the Benjamini-Hochberg Procedure (BHP) to control false discovery rate. In the Method section (statistical analysis part) the following explanation is included:

Multiple testing corrections were performed using Benjamini–Hochberg procedure to control the false discovery rate and only those p values that passed the criteria posted by the procedure were considered significant. Controlling false discovery rate has been recognized as appropriate when the goal is to identify candidate effects from a large set and recognize groups that can be tested more rigorously in follow-up confirmatory experiments [36].

  1. Cramer, A. O., van Ravenzwaaij, D., Matzke, D., Steingroever, H., Wetzels, R., Grasman, R. P., Waldorp, L. J., & Wagenmakers, E. J. (2016). Hidden multiplicity in exploratory multiway ANOVA: Prevalence and remedies. Psychonomic bulletin & review23(2), 640–647. https://doi.org/10.3758/s13423-015-0913-5

Specifically for Figure 2, using the formula (i/m)*Q, where = rank of p-value , = 10 and Q = chosen false discovery rate (0.05) the condition appeared that only p values below 0.005 could be considered significant (as you already noted in the comment), so all obtained p values (0.027) become insignificant. The same is with the rest of the comparisons related Figure 2.

Please note that throughout the study results are subject to BHP because, as you said in the comments below, the way the study compares the results, using multiple comparisons, gives rise to this.

Reviewer 1, comment 4. Figure 4 suffers from the same issue: the authors seem to liberally utilize the Mann-Whitney U test (which is meant to do a one-time comparison of two populations) to compare 5 groups against the control and each other (20 different permutations?). This statistical method is woefully inadequate for this task. I agree that there may be some differences and trends here, but the statistics must be adjusted for the multiple comparisons used here.

Response to Reviewer 1, comment 4. Thank you for the comment, it is generally correct. However, there were no 20 comparisons but 4 mandatory, i.e. control group versus sucrose groups (32%, 16%, 8% and 4%) and, if H statistics allow, additional comparisons between sucrose groups (6 comparisons: 32% vs. 16%, 32% vs. 8%, 32% vs 4%, 16% vs. 8%, 16% vs. 4% and 8% vs. 4%). As we indicated in the response to your comment 3, throughout the study results are subject to the correction using the Benjamini–Hochberg procedure to control the false discovery rate, which is appropriate (accentuated in the Statistical analysis subsection) when the goal is to identify candidate effects from a large set and recognize groups that can be tested more rigorously in follow-up confirmatory experiments.

Reviewer 1, comment 5. Claiming that downshift may be an important factor for locomotion is a bit misleading: 16% and 8% downshift is no different from control, only different from 4% downshift? How is this meaningful is unclear. I recommend that the authors clarify that there is no difference to control, I am not convinced the measured difference is not simply due to under sampling.

Response to Reviewer 1, comment 5. Thank you for the comment. After applying the Benjamini–Hochberg procedure this significances were lost and we stated in the results section (changes in red):

 Locomotor activity (Figure 4A) was highly similar in the control and sucrose exposed groups. Sucrose downshift appeared as an important factor for locomotor activity (H (3, N = 24) = 7.76, p = 0.05), but post-hoc analysis followed by the Benjamini–Hochberg procedure did not reveal significant differences.

Reviewer 1, comment 6. One cannot help but wonder if the lack of effects across the paper are due to the relatively short exposure to sucrose. In humans, 10 days of sugar consumption is not likely to have any measurable effect either. A more appropriate timescale for this experiment would be at least a couple of months on high sugar diet.

Response to Reviewer 1, comment 6. Thank you for the comment. As we already indicated in the response to your comment 1, this issue prompted us to add some explanations of the methodological design, i.e. explanation for the choice of treatment length. Please find the following added material:

An important variable to consider is the duration of sucrose exposure. Defining the length of the experimental treatment largely depends on what the study wants to model (specific conditions in one segment of a certain life stage or in the whole life stage or in the whole life span). In the present study, we attempted to model several-years-long unlimited sugar consumption during adulthood. In order to adequately set period of the exposure we considered that one day in the life of a human and a mouse do not mean the same as mice have a diminutive lifespan compared to humans and, during the adult phase, 2.60 mice days are equivalent to one human year (for a detailed explanation see [55]). Accordingly, 10 days in the life of an adult mouse should be seen as about 3.8 years of the life of an adult human.

  1. Dutta, S., & Sengupta, P. (2016). Men and mice: Relating their ages. Life sciences152, 244–248. https://doi.org/10.1016/j.lfs.2015.10.025

Reviewer 1, comment 7. The authors make a lot of claims about how their animal model (both the sucrose consumption and the behavioral paradigms) is related to various human conditions. I recommend dialing this rhetoric back a bit as mouse behavior in an open field or in an EPM is difficult to directly relate to human psychology.

Response to Reviewer 1, comment 7. Thank you for the comment. We checked how many times the word human appears in the manuscript and found it 4 times (in the initial version of the manuscript), in the following context:

Introduction, lines 42-46: Consequently, unlike metabolic and obesity-related health disparities, behavioral outcomes of unlimited sucrose intake (USI) outside the food context did not get much attention in experimental research, even though this manner of sugar intake is a more valid representation of human daily exposure to sucrose.

Introduction, lines 93-97: To provide a model that would incorporate human-like patterns of self-administration in laboratory animals, we opted for a design which allows free choice access and thus applied the two-bottle choice paradigm (2BCP) in male C57BL/6J mice, allowing them unlimited access to the potent sucrose solution (32%) in home cages, without affecting social environment/hierarchy.

References:

Flora, S. R., & Polenick, C. A. (2013). Effects of sugar consumption on human behavior and performance. The Psychological Record, 63(3), 513–524. https://doi.org/10.11133/j.tpr.2013.63.3.008

Ferreira, S. O. (2019). Emotional activation in human beings: procedures for experimental stress induction. Psicologia USP, 30. https://doi.org/10.1590/0103-6564e20180176

In the revised manuscript, the word human appears 4 more times in an added paragraph explaining the choice of treatment length. We didn't find the critical points you're talking about in terms of making a lot of claims about how the tested animal model (both sucrose consumption and behavioral paradigms) relates to various human conditions. Everything about the potential connection between sugar intake and human behavior is given in the introductory part, as an explanation of why this topic is worth an experimental approach, while the discussion is based on the obtained results  (for example, the first paragraph in the discussion section: Increased sugar consumption becomes an important public health concern and there is a need to properly understand consequences. The present study shows, in the mouse model, that USI...). According to your recommendation...I recommend dialing this rhetoric back a bit as mouse behavior in an open field or in an EPM is difficult to directly relate to human psychology... please explicitly state which parts of the text we should pay attention to. we reviewed the text in detail but did not come across any places that overemphasize the importance obtained results.

Minor problems

Reviewer 1, comment 8. References in the Introduction are rather scarce. There are some big claims made in the Introduction related to a variety of things ranging from obesity, sugar intake and ADHD, attention, and other behaviors, all with a single citation. Since these are meant to motivate the study, extensive citations are needed to showcase the robustness of the authors’ claims.

Response to Reviewer 1, comment 8. Thank you for the comment and suggestion, new references have been added.

  1. Nigg, J. T., Johnstone, J. M., Musser, E. D., Long, H. G., Willoughby, M. T., & Shannon, J. (2016). Attention-deficit/hyperactivity disorder (ADHD) and being overweight/obesity: New data and meta-analysis. Clinical psychology review43, 67–79. https://doi.org/10.1016/j.cpr.2015.11.005
  2. Weissenberger, S., Ptacek, R., Vnukova, M., Raboch, J., Klicperova-Baker, M., Domkarova, L., & Goetz, M. (2018). ADHD and lifestyle habits in Czech adults, a national sample. Neuropsychiatric disease and treatment14, 293–299. https://doi.org/10.2147/NDT.S148921
  3. Johnson, R. J., Gold, M. S., Johnson, D. R., Ishimoto, T., Lanaspa, M. A., Zahniser, N. R., & Avena, N. M. (2011). Attention-deficit/hyperactivity disorder: is it time to reappraise the role of sugar consumption?. Postgraduate medicine123(5), 39–49. https://doi.org/10.3810/pgm.2011.09.2458

Reviewer 1, comment 9. Line 115-116: take out “the most common 115 inbred mouse strain employed in biomedical research”, this appears to be the authors’ opinion, not substantiated by anything.

Response to Reviewer 1, comment 9. Thank you for the comment. The reference that highlights this fact was omited by mistake and it is listed in the revised form of the manuscript.

  1. Sloin, H. E., Bikovski, L., Levi, A., Amber-Vitos, O., Katz, T., Spivak, L., Someck, S., Gattegno, R., Sivroni, S., Sjulson, L., & Stark, E. (2022). Hybrid Offspring of C57BL/6J Mice Exhibit Improved Properties for Neurobehavioral Research. eNeuro9(4), ENEURO.0221-22.2022. https://doi.org/10.1523/ENEURO.0221-22.2022

Reviewer 1, comment 10. Figure 2: data missing. I understand that 2 cages were taken out of the study due to not consuming sucrose during the downshift but the way the Methods are written suggests that they were only taken out from downshift and after. Yet, I only see 6 data points in figure 2 instead of 8. Why are there data points missing?

Response to Reviewer 1, comment 10. Thank you for the comment, we made clarifications. In the first paragraph of the results section we included explanation:

Please note that only cages with animals that passed all exclusion criteria (and animals contributed to the behavioral results) were taken into consideration. Therefore, the number of cages is N = 6 (as 2 cages were excluded; for details please see subsection 2.6.2. Exclusion criteria during the experimental phase 1 and experimental phase 2).

Reviewer 2 Report

In my opinion, the manuscript prepared by Olga Dubljević it is a well-conducted study in a restrictive area of research. 

To improve the quality of the manuscript I recommend only minor comments:

- The research base should be added by the authors in the Abstract.

- Kindly revise the figure 6 legend. It is incomplete.

- Please list, in the same manner, all the Figures in the main text (please see line 412).

Author Response

Reviewer  2

In my opinion, the manuscript prepared by Olga Dubljević it is a well-conducted study in a restrictive area of research. 

To improve the quality of the manuscript I recommend only minor comments:

Reviewer 2, comment 1. The research base should be added by the authors in the Abstract.

Response to Reviewer 2, comment 1. Thank you for the comment/suggestion. The first sentence in the Abstract section has been changed to indicate the theoretical frame on which the study is based.

Reviewer 2, comment 2. Kindly revise the figure 6 legend. It is incomplete.

Response to Reviewer 2, comment 2. Thank you for the comment/suggestion. The figure 6 legend is completed.

Reviewer 2, comment 3. Please list, in the same manner, all the Figures in the main text (please see line 412).

Response to Reviewer 2, comment 3. Thank you for the comment, corrections were made.

Round 2

Reviewer 1 Report

The authors made a good faith effort to improve their manuscript. I appreciate the improved statistical testing. I also understand that repeating a full study or significantly increasing the sample size is not always feasible.